# Arachidonic Acid Evokes an Increase in Intracellular Ca^2+^ Concentration and Nitric Oxide Production in Endothelial Cells from Human Brain Microcirculation

**DOI:** 10.3390/cells8070689

**Published:** 2019-07-09

**Authors:** Roberto Berra-Romani, Pawan Faris, Sharon Negri, Laura Botta, Tullio Genova, Francesco Moccia

**Affiliations:** 1Biomedicine School, Benemerita Universidad Autonoma de Puebla, 72000 Puebla, Mexico; 2Laboratory of General Physiology, Department of Biology and Biotechnology “L. Spallanzani”, University of Pavia, 27100 Pavia, Italy; 3Research Center, Salahaddin University, Erbil 44001, Iraq; 4Department of Life Sciences and Systems Biology, University of Torino, 10123 Torino, Italy; 5Department of Surgical Sciences, University of Torino, 10126 Torino, Italy

**Keywords:** arachidonic acid, brain microvascular endothelial cells, neurovascular coupling, cerebral blood flow, Ca^2+^ signalling, nitric oxide, inositol-1,4,5-trisphosphate receptors, two-pore channels 1-2, transient receptor potential vanilloid 4

## Abstract

It has long been known that the conditionally essential polyunsaturated arachidonic acid (AA) regulates cerebral blood flow (CBF) through its metabolites prostaglandin E2 and epoxyeicosatrienoic acid, which act on vascular smooth muscle cells and pericytes to vasorelax cerebral microvessels. However, AA may also elicit endothelial nitric oxide (NO) release through an increase in intracellular Ca^2+^ concentration ([Ca^2+^]_i_). Herein, we adopted Ca^2+^ and NO imaging, combined with immunoblotting, to assess whether AA induces intracellular Ca^2+^ signals and NO release in the human brain microvascular endothelial cell line hCMEC/D3. AA caused a dose-dependent increase in [Ca^2+^]_i_ that was mimicked by the not-metabolizable analogue, eicosatetraynoic acid. The Ca^2+^ response to AA was patterned by endoplasmic reticulum Ca^2+^ release through type 3 inositol-1,4,5-trisphosphate receptors, lysosomal Ca^2+^ mobilization through two-pore channels 1 and 2 (TPC1-2), and extracellular Ca^2+^ influx through transient receptor potential vanilloid 4 (TRPV4). In addition, AA-evoked Ca^2+^ signals resulted in robust NO release, but this signal was considerably delayed as compared to the accompanying Ca^2+^ wave and was essentially mediated by TPC1-2 and TRPV4. Overall, these data provide the first evidence that AA elicits Ca^2+^-dependent NO release from a human cerebrovascular endothelial cell line, but they seemingly rule out the possibility that this NO signal could acutely modulate neurovascular coupling.

## 1. Introduction

The 20-carbon omega-6 polyunsaturated fatty acid (PUFA) arachidonic acid (AA) is incorporated in the plasma membrane esterified to phospholipids at the second alcohol of glycerol in the sn-2 position and exerts multiple homeostatic, structural, and signalling functions [1]. AA is released from the membrane pool by the activation of three distinct phospholipases, A2, C, and D, upon cell stimulation to exert either an autocrine and/or a paracrine effect on adjoining cells [1,2]. For instance, synaptically released glutamate activates *N*-methyl-d-aspartate receptors (NMDARs), thereby raising intracellular Ca^2+^ levels within the dendritic tree and recruiting the Ca^2+^-dependent enzyme phospholipase A2 (PLA2). Neuronal PLA2, in turn, mobilizes AA from membrane phospholipids; hereafter, AA is converted by cyclooxygenase-2 (COX-2) into prostaglandin E2 (PGE2) to relax cerebral microvessels [3]. Likewise, neuronal activity causes an increase in astrocytic Ca^2+^ concentration, which also recruits PLA2 to stimulate the synthesis of a number of AA-derived metabolites that serve as vasodilatory mediators. These include PGE2 and epoxyeicosatrienoic acids (EETs) that are produced by COX-1 and cytochrome P450 epoxygenase [2,4], respectively. Alternately, astrocyte-derived AA is released onto neighbouring vascular smooth muscle cells (VSMCs) or pericytes where it can be metabolized into the vasoconstrictor 20-hydroxyeicosatetraenoic acid (20-HETE) by cytochrome P450 4A2 [2,4,5,6]. AA is thus regarded as a key regulator of neurovascular coupling (NVC), the mechanism whereby neuronal activity results in the elevation of local cerebral blood flow (CBF), which is indispensable to maintain proper supply of oxygen and nutrients to active brain regions [2,3,4,7]. For instance, the hemodynamic response to whisker stimulation in the somatosensory cortex was dramatically impaired by inhibiting the production of PGE2 and EETs by, respectively, pyramidal neurons and astrocytes [8]. Conversely, only EETs were responsible for the elevation in CBF evoked by stimulation of basal forebrain cholinergic neurons [9]. On the other hand, 20-HETE has been shown to induce capillary vasoconstriction in both the molecular [10] and granular [5] layer of the cerebellum, depending on the presence of nitric oxide (NO), that suppresses cytochrome P450 4A2 activity [2].

A recent series of studies demonstrated that cerebral endothelial cells, which line the lumen of brain microvessels and are in close apposition to both astrocyte end-feet and neuronal projections within the neurovascular unit, may also translate neuronal activity into NO-dependent vasodilation [11,12,13]. For instance, the neurotransmitter acetylcholine triggers a biphasic increase in intracellular Ca^2+^ concentration ([Ca^2+^]_i_) which leads to the recruitment of Ca^2+^/calmodulin (CaM)-sensitive endothelial NO synthase (eNOS) in hCMEC/D3 cells [14], which are a widely utilized human brain microvascular endothelial cell line [15]. Likewise, acetylcholine and glutamate were found to induce repetitive oscillations in [Ca^2+^]_i_ and NO production in mouse brain microvascular endothelial cells [14,16]. The Ca^2+^ response to extracellular neurotransmitters in cerebrovascular endothelial cells is mainly mediated by inositol-1,4,5-trisphosphate (InsP_3_) receptors (InsP_3_Rs) and two-pore channels 1 and 2 (TPC1-2), which mediate Ca^2+^ release from endoplasmic reticulum (ER) and lysosomes, respectively, and by store-operated Ca^2+^ entry (SOCE), a widespread Ca^2+^-permeable route activated following depletion of the ER Ca^2+^ reservoir [14,16,17]. Intriguingly, it has long been known that AA directly induces Ca^2+^-dependent NO release in vascular endothelial cells [18,19,20,21] and endothelial colony-forming cells (ECFCs) [22]. The Ca^2+^ response to AA is shaped by multiple components of the endothelial Ca^2+^ toolkit, including InsP_3_Rs and TPC1, which mediate endogenous Ca^2+^ release, respectively, and transient receptor potential vanilloid 4 (TRPV4), which mediates extracellular Ca^2+^ influx [18,19,20,21,23]. Conversely, AA is seemingly unable to activate SOCE [24,25,26]. A recent investigation suggested that AA may be converted by cytochrome P450 2U1 epoxygenase into 20-HETE also in hCMEC/D3 cells and rat brain microvessels [27]. Furthermore, a preliminary report showed that AA may trigger intracellular Ca^2+^ signals in primary human brain microvascular endothelial cells, but the underlying signalling pathway was not clearly elucidated and their ability to promote NO was not evaluated [23].

Herein, we investigated for the first time whether and how exogenous AA induces intracellular Ca^2+^ signalling and NO release in hCMEC/D3 cells. AA elicited a biphasic increase in [Ca^2+^]_i_ which was shaped by endogenous Ca^2+^ release through type 3 inositol-1,4,5-trisphosphate receptor isoform (InsP_3_R3) and two-pore channels 1 and 2 (TPC1-2) and extracellular Ca^2+^ influx through TRPV4. We further demonstrated that AA induced robust Ca^2+^-dependent NO release. However, AA-induced NO production was remarkably delayed as compared to intracellular Ca^2+^ signalling. In addition, AA-induced NO release was mainly dependent upon TPC1-2 and TRPV4 activation rather than InsP_3_R3. Overall, these data provide the evidence that AA is able to induce intracellular Ca^2+^ signalling and NO release in human brain microvascular endothelial cells, although this NO release is seemingly too slow to account for the vasorelaxing effects of AA observed in vivo.

## 2. Materials and Methods

### 2.1. Cell Culture

Human cerebral microvascular endothelial cells (hCMEC/D3) were obtained from Institut National de la Santé et de la Recherche Médicale (INSERM, Paris, France) and cultured as extensively illustrated elsewhere [17]. Only hCMEC/D3 cells between passage 25 and 35 were employed in the present investigation. The cells were seeded at a concentration of 27,000 cells/cm^2^ and grown in tissue culture flasks coated with 0.1 mg/mL rat tail collagen type 1, in a medium having the following composition: EBM-2 medium (Lonza, Basel, Switzerland) supplemented with 5% foetal bovine serum, 1 ng/mL basic fibroblast growth factor, 1% penicillin–streptomycin, 5 μg/mL ascorbic acid, 1.4 μM hydrocortisone, 1/100 chemically defined lipid concentrate (Life Technologies, Milan, Italy), and 10 mM 4-(2-hydroxyethyl)-1-piperazineethanesulfonic acid (HEPES). The cells were cultured at 37 °C, 5% CO_2_ saturated humidity.

### 2.2. Solutions

The composition of the physiological salt solution (PSS) used to load the cells with the Ca^2+^- and NO-sensitive dyes and to bath the cells during recordings had the following composition: 150 mM NaCl, 6 mM KCl, 1.5 mM CaCl_2_, 1 mM MgCl_2_, 10 mM glucose, and 10 mM HEPES. In order to obtain an extracellular solution devoid of Ca^2+^ (0Ca^2+^), Ca^2+^ was substituted with 2 mM NaCl and 0.5 mM EGTA was added. Solutions were titrated to pH 7.4 with NaOH. The osmolality of PSS was measured with an osmometer (Wescor 5500, Wescor, Logan, UT, USA) and resulted to be 300–310 mOsm/L.

### 2.3. [Ca^2+^]_i_ and NO Imaging

Ca^2+^ imaging was carried out as described elsewhere [17]. Briefly, hCMEC/D3 cells were loaded with 4 µM Fura-2 acetoxymethyl ester (Fura-2/AM; 1 mM stock in dimethyl sulfoxide) in PSS. The cells were maintained in the presence of Fura-2 for 30 min at 37 °C and 5% CO_2_ saturated humidity. After de-esterification in PSS, the coverslip (8 mm) was mounted on the bottom of a Petri dish and the cells observed by an upright epifluorescence Axiolab microscope (Carl Zeiss, Oberkochen, Germany) equipped with a Zeiss ×40 Achroplan objective (water immersion, 2.0 mm working distance, 0.9 numerical aperture). The cells were excited alternately at 340 and 380 nm by using a filter wheel (Lambda 10, Sutter Instrument, Novato, CA, USA). The emitted fluorescence was detected at 510 nm by using an Extended-ISIS CCD camera (Photonic Science, Millham, UK). Custom-made software, working in the LINUX environment, was used to drive the CCD camera and the filter wheel, and to measure and plot on-line the fluorescence from 15 to 25 rectangular “regions of interest” (ROI), each corresponding to well defined single cells. [Ca^2+^]_i_ was monitored by measuring, for each ROI, the ratio of the mean fluorescence emitted at 510 nm when exciting alternatively at 340 and 380 nm (Ratio (F_340_/F_380_)). An increase in [Ca^2+^]_i_ causes an increase in the ratio [17]. Ratio measurements were performed and plotted on-line every 3 s. The experiments were performed at room temperature (22 °C), as discussed in [28].

NO was measured as described in [17]. Briefly, hCMEC/D3 cells were loaded with the membrane permeable NO-sensitive dye 4-amino-5-methylamino-2’,7’-difluorofluorescein (DAF-FM) diacetate (10 µM) in PSS. The cells were maintained in the presence of DAF-FM for 1 h at 37 °C and 5% CO_2_ saturated humidity. Subsequently, the dye was de-esterified in PSS for a further hour. DAF-FM fluorescence was measured by using the same equipment described for Ca^2+^ recordings but with a different filter set, i.e., excitation at 480 nm and emission at 535 nm wavelength (emission intensity was shortly termed “NO_i_”). The changes in DAF-FM fluorescence induced by AA were recorded and plotted on-line every 5 s. Again, off-line analysis was performed by using custom-made macros developed by Microsoft Office Excel software. The experiments were performed at room temperature (22 °C).

### 2.4. Western Blotting

Semi-confluent adherent cells were carefully lysate by using ice-cold RIPA buffer (150 mM NaCl, 25 mM Tris-HCl pH 7.6, 1% sodium deoxycholate, 1% NP-40, and 0.1% SDS) adding protease inhibitor cocktail (Sigma-Aldrich, Milan, Italy). In order to determine the protein concentration, a bicinchoninic acid kit (Thermo Fisher Scientific, Waltham, MA, USA) was used following the manufacturer’s instructions. Antibodies against TRPV4 (ab94868; Abcam, Cambridge, UK) and antibodies anti–β-actin (Sigma-Aldrich) were used as indicated by the manufacturer. Chemiluminescence assays were conducted by using western blotting plus ECL (Perkin Elmer, Milano, Italy).

### 2.5. Statistical Analysis

All the data were obtained from hCMEC/D3 cells from no fewer than three independent experiments. The peak amplitude of AA-evoked Ca^2+^ signals was measured as the difference between the ratio at the Ca^2+^ peak and the mean ratio of 30 sec baseline before the peak. The dose–response relationship was fitted by using the following equation:(1)$$Y\;=\;\frac{100}{1+\frac{EC_{50}}{\left[\mathrm{AA}\right]}}$$
where Y is the amplitude of the Ca^2+^ response, [AA] is the AA concentration, and EC_50_ is the half-maximal effective concentration, i.e., the [AA] that elicited 50% of the maximal response.

The peak amplitude of AA-induced NO production was evaluated as the difference between the maximal increase in DAF-FM fluorescence and the average of 1 min baseline before the peak. Pooled data are given as average ± standard error (SE) and statistical significance (*p*  < 0.05) was evaluated by two-tailed Student’s *t* test for unpaired observations or one-way ANOVA analysis followed by the post-hoc Dunnett’s test as appropriate. Data relative to both Ca^2+^ and NO signals are summarized as mean  ±  SE, whereas the number of cells analysed for each condition is indicated in the corresponding bar histograms.

### 2.6. Chemicals

Fura-2/AM and DAF-FM were purchased from Molecular Probes (Molecular Probes Europe BV, Leiden, The Netherlands). GSK1016790A (GSK) was obtained from Tocris (Bristol, UK). All the chemicals were of analytical grade and obtained from Sigma Chemical Co. (St. Louis, MO, USA).

## 3. Results

### 3.1. AA Triggers an Increase in [Ca^2+^]_i_ in hCMEC/D3 Cells

In order to assess whether AA induces intracellular Ca^2+^ signals, hCMEC/D3 cells were loaded with the Ca^2+^-sensitive fluorophore, Fura-2, as described in paragraph 2.3. Neuronal activity leads to remarkable increase in synaptic AA concentration, which can rise up to 50–200 μM [29,30,31]. Accordingly, AA induced an increase in [Ca^2+^]_i_ in hCMEC/D3 cells that started at 1 μM (Figure 1A), reached the maximal response at 300 μM, and whose dose–response relationship could be fitted by a sigmoidal curve with an EC_50_ of 8.4 μM (Figure 1B). A careful inspection of the Ca^2+^ traces revealed that 3 μM AA elicited a slow elevation in [Ca^2+^]_i_ that remained stable during the continuous presence of the agonist in the bath (Figure 1A). Conversely, when applied at 30 μM and 300 μM, AA induced a rapid Ca^2+^ transient that slowly decayed to a sustained plateau of elevated [Ca^2+^]_i_ (Figure 1A), as recently shown in ECFCs [22].

As synaptic AA concentration may achieve the mid-to-high micromolar range [29,30,31], we used 30 μM AA throughout the remainder of the study. The Ca^2+^ response to AA (30 μM) was reversible as, when AA was removed from the bath at the peak of the Ca^2+^ rise, the [Ca^2+^]_i_ rapidly returned to the baseline, whereas the subsequent addition of AA raised intracellular Ca^2+^ levels again (Figure 1C). AA-induced endothelial Ca^2+^ signals may be triggered by many of its derivatives, such as 5,6- epoxyeicosatrienoic (5,6-EET) and 8,9-EET and leukotriene C4, which are synthesized, respectively, by cytochrome P450 epoxygenase and glutathione S-transferase 2 [32,33,34,35,36]. Therefore, we evaluated the Ca^2+^ response to the non-metabolized AA analogue, eicosatetraynoic acid (ETYA), as described in other types of endothelial cells [22,37]. As shown in Figure 2, ETYA (30 μM) induced an increase in [Ca^2+^]_i_ whose kinetics and amplitude were similar to those of the Ca^2+^ signal induced by AA (30 μM) in the same batches of hCMEC/D3 cells (Figure 2A,B). Taken together, these findings demonstrate that AA has no need to be metabolized to induce intracellular Ca^2+^ signals in human brain microvascular endothelial cells.

### 3.2. AA-Induced Intracellular Ca^2+^ Signals Required Endogenous Ca^2+^ Release and Extracellular Ca^2+^ Influx

As mentioned earlier, AA-induced endothelial Ca^2+^ signals impinge on endogenous Ca^2+^ release and extracellular Ca^2+^ influx [18,19,20,21]. In order to address this issue, we evaluated the Ca^2+^ response to AA in hCMEC/D3 cells in the absence of extracellular Ca^2+^ (0Ca^2+^), after which extracellular Ca^2+^ was restituted while maintaining the agonist in the perfusate. Under 0Ca^2+^ conditions, 30 μM AA induced a transient elevation in [Ca^2+^]_i_, which was due to Ca^2+^ mobilization from the intracellular stores, whereas reintroduction of external Ca^2+^ caused a second intracellular Ca^2+^ wave (Figure 3A), which was sustained by extracellular Ca^2+^ influx [22].

The same results were obtained when AA concentration was raised up to 50 μM (Figure 3B). As expected [22], endogenous Ca^2+^ release and extracellular Ca^2+^ influx were significantly (*p* < 0.05) higher at the highest AA concentration (Figure 3C). Overall, these findings demonstrate that intracellular Ca^2+^ release and extracellular Ca^2+^ influx mediate AA-evoked Ca^2+^ signals in hCMEC/D3 cells.

### 3.3. InsP_3_R3 and TPC1-2 Mediate AA-Induced Endogenous Ca^2+^ Release

Recent studies demonstrated that InsP_3_Rs and TPC1-2 mediate AA-induced endogenous Ca^2+^ release in human ECFCs [22] and rat pancreatic β cells [38]. Of note, a thorough characterization of their Ca^2+^ toolkit disclosed that hCMEC/D3 cells express only one of the three InsP_3_R isoforms, i.e. InsP_3_R3, and TPC1-2, while they lack InsP_3_R1-2 and RyR1-3 [17]. We first focused on the contribution of ER Ca^2+^ release through InsP_3_R3 to AA-induced intracellular Ca^2+^ release. As shown in Figure 4A, the endogenous Ca^2+^ response to AA (30 μM) was abrogated by 2-aminoethoxydiphenyl borate (2-APB) (50 μM, 20 min) (Figure 4B), which inhibits InsP_3_Rs under 0Ca^2+^ conditions [17,39], and by depletion of the ER Ca^2+^ content with cyclopiazonic acid (CPA) (10 μM) (Figure 4C). As previously described [17], CPA caused a transient elevation in [Ca^2+^]_i_ by unmasking the passive ER Ca^2+^ leak followed by Ca^2+^ extrusion across the plasma membrane under 0Ca^2+^ conditions (Figure 4C). The following addition of AA (30 μM) under these conditions failed to increase the [Ca^2+^]_i_, thereby confirming the role of ER Ca^2+^ in AA-evoked intracellular Ca^2+^ response. 

Next, we evaluated the contribution of lysosomal Ca^2+^ release through TPC1-2. The Ca^2+^ response to AA (30 μM) was prevented upon disruption of the lysosomal Ca^2+^ pool with glycyl-L-phenylalanine 2-naphthylamide (GPN) (200 μM) administered under 0Ca^2+^ conditions (Figure 4D). In addition, AA-evoked intracellular Ca^2+^ mobilization was suppressed by NED-19 (100 µM, 30 min) (Figure 4E), which selectively blocks TPC1-2 [17,40]. The statistical analysis of these results is shown in Figure 4F. In aggregate, these data provide the evidence that AA-evoked intracellular Ca^2+^ mobilization in hCMEC/D3 cells is mediated by InsP_3_R3 and TPC1-2.

### 3.4. TRPV4 is Expressed and Mediates AA-Evoked Extracellular Ca^2+^ Influx

TRPV4 is virtually the only Ca^2+^-permeable pathway responsible for AA-induced extracellular Ca^2+^ influx in endothelial cells [22,37,43,44]. Therefore, we first assessed whether a functional TRPV4 protein was expressed in hCMEC/D3 cells. We carried out a Western Blot analysis by exploiting an affinity-purified antibody, which detected a major band at 91 kDa (Figure 5A), the expected molecular weight for TRPV4 [45]. Thereafter, we exposed hCMEC/D3 cells to GSK (20 nM), a specific TRPV4 agonist [45,46]. GSK (20 nM) caused a robust increase in [Ca^2+^]_i_ in the presence, but not in the absence, of extracellular Ca^2+^ (Figure 5B), which is consistent with the notion that TRPV4 mediates extracellular Ca^2+^ influx. Notably, GSK-induced Ca^2+^ influx was suppressed by RN-1734 (20 μM, 20 min), a selective TRPV4 blocker [45,47], in the majority of the cells, whereas it was remarkably inhibited in the remaining (Figure 5C,D). These preliminary experiments provide the evidence that TRPV4 is expressed and mediates extracellular Ca^2+^ influx in hCMEC/D3 cells. 

Next, we explored the role of TRPV4 in AA-evoked intracellular Ca^2+^ signals. In agreement with previous reports in endothelial cells [22,37,43,44], the pharmacological blockade of TRPV4 with RN-1734 (20 μM, 20 min) decreased the magnitude and curtailed the duration of AA-evoked Ca^2+^ response (Figure 6A,B). To further confirm this observation, we assessed whether AA could increase the [Ca^2+^]_i_ after exhaustion of the ER Ca^2+^ pool and full SOCE activation with CPA (10 μM) in the presence of extracellular Ca^2+^. Accordingly, SOCE is elicited by pharmacological depletion of the ER Ca^2+^ reservoir with CPA and can no longer be activated if ER cisternae are not replenished with Ca^2+^ because of CPA-dependent SERCA inhibition [17,48]. Therefore, under these conditions, any extracellular stimulation will be able to increase the [Ca^2+^]_i_ only by mainly acting on store-independent Ca^2+^-permeable channels, such as TRPV4, located on the plasma membrane [22,24]. Accordingly, previous experiments revealed that AA did not evoke any Ca^2+^ signal under 0Ca^2+^ conditions and following treatment with CPA (Figure 4C). Indeed, when AA (30 μM) was added upon SOCE activation with CPA (10 μM) for 30 min (early response to CPA not shown) in the presence of extracellular Ca^2+^, it elicited a sustained elevation in [Ca^2+^]_i_ that was suppressed by RN-1734 (20 μM, 20 min) (Figure 6C,D). Therefore, these findings clearly show that TRPV4 mediates AA-induced Ca^2+^ influx in hCMEC/D3 cells.

### 3.5. AA-Evoked Intracellular Ca^2+^ Signaling Drives NO Release in hCMEC/D3 Cells

In order to assess whether AA-evoked Ca^2+^ signalling results in NO release, hCMEC/D3 cells were loaded with the NO-sensitive dye, DAF-FM, as illustrated in Section 2.3. Unlike the neurotransmitter acetylcholine [17], AA (30 μM) elicited a delayed increase in DAF-FM fluorescence that significantly lagged behind the onset of the Ca^2+^ signal (Figure 7A): accordingly, the NO signal arose with a latency of 149.4 ± 6.2 sec (*n* = 368) and slowly increased with a time-to-peak of 1230 ± 39 sec (*n* = 339). AA-elicited NO release was abrogated by L-N^G^-Nitro-L-arginine methyl ester (L-NAME) (100 µM, 1 h) (Figure 7B), a widely employed NOS inhibitor [22,49], or 1,2-Bis(2-aminophenoxy)ethane-N,N,N′,N′-tetraacetic acid tetrakis(acetoxymethyl ester) (BAPTA) (30 µM, 2 h) (Figure 7A), a membrane-permeant buffer of Ca^2+^ levels [22,49]. Notably, ETYA (30 μM) induced an increase in DAF-FM fluorescence whose kinetics and magnitude were again overlapping with those of the signal elicited by AA (30 μM) (Figure 7B). These data, therefore, indicate that AA-evoked Ca^2+^ signalling results in NO release also in hCMEC/D3 cells. 

In order to dissect the contribution of extra- and intracellular Ca^2+^ sources to NO production, hCMEC/D3 cells were challenged with AA (30 μM) under 0Ca^2+^ conditions. Removal of external Ca^2+^ significantly (*p* < 0.05) reduced AA-elicited NO release, whereas restoration of extracellular Ca^2+^ levels caused a further elevation in DAF-FM fluorescence (Figure 8B). Accordingly, also the pharmacological blockage of TRPV4 with RN-1734 (20 μM, 30 min) remarkably impaired AA-elicited NO synthesis (Figure 8C). We adopted the same pharmacological strategy as that described in Section 3.3 to dissect the contribution of the ER and lysosomal Ca^2+^ stores to AA-induced NO release. Surprisingly, inhibition of InsP_3_R3 with 2-APB (50 μM, 20 min) (Figure 8D) and depletion of the ER Ca^2+^ store with CPA (10 μM, 30 min) (Figure 8E) did not affect the NO signal. Conversely, preventing lysosomal Ca^2+^ release through TPC1-2 with GPN (200 µM, 30 min) (Figure 8F) and NED-19 (100 µM, 30 min) (Figure 8G) abolished AA-induced NO release. Altogether, these findings strongly suggest that TPC1-2 and TRPV4 are mainly responsible for the Ca^2+^-dependent recruitment of eNOS induced by AA in hCMEC/D3 cells. However, the slow kinetics of NO release are not compatible with a direct involvement of AA in NVC.

## 4. Discussion

Herein, we demonstrate for the first time that the conditionally essential PUFA AA is able to elicit NO release from a human brain microvascular endothelial cell line in a Ca^2+^-dependent manner. The Ca^2+^ response to AA involves endogenous Ca^2+^ release from the ER and lysosomal Ca^2+^ pools through, respectively, InsP_3_R3 and TPC1-2 and extracellular Ca^2+^ influx through TRPV4. However, only TPC1-2 channels and TRPV4 are seemingly coupled to eNOS activation and NO release. AA is the precursor of multiple vasoactive mediators shaping the hemodynamic response to neuronal activity. However, the slow kinetics of AA-evoked NO release make it unlikely that this signal directly contributes to NVC.

### 4.1. Intracellular Ca^2+^ Signals Induced by AA in hCMEC/D3 Cells

It has long been known that synaptic activity leads to a significant increase in the local AA concentration, which can raise up to 50–200 μM following the Ca^2+^-dependent recruitment of PLA2 in both postsynaptic neurons and perisynaptic astrocytes [29,30,31]. AA, in turn, regulates multiple synaptic processes, including neurotransmitter release [50], long-term potentiation [29,30], and local increase in CBF [2,7]. Intriguingly, a recent investigation demonstrated that 50 μM AA could elicit an increase in [Ca^2+^]_i_ in primary human brain endothelial cells [23]. As endothelial Ca^2+^ signals are emerging as an unsuspected regulator of NVC [11,12,14,17,51] and AA may directly trigger Ca^2+^-dependent NO release in other endothelial cell types [18,19,20,21,22], we sought to assess whether it could elicit intracellular Ca^2+^ signals and NO release also in human cerebrovascular endothelial cells [15]. 

### 4.2. The Dose–Response Relationship of AA-Evoked Intracellular Ca^2+^ Signals in hCMEC/D3 Cells

We found that AA induced a dose-dependent increase in [Ca^2+^]_i_ within the physiological concentration range that it reaches during synaptic activity. The EC_50_ of AA-evoked Ca^2+^ signal was indeed 8.4 μM, whereas the peak Ca^2+^ response was attained at 300 μM. A similar analysis revealed that breast tumor-derived endothelial cells (b-TECs) displayed a higher sensitivity as compared to hCMEC/D3 cells, as the Ca^2+^ response to AA arose at 0.5 μM and achieved its peak at 10 μM [37]. However, AA-evoked Ca^2+^ signals in ECFCs appeared at concentrations higher than 2 μM and attained the maximal amplitude at 100 μM [22,45]. Likewise, a sizeable elevation in [Ca^2+^]_i_ was detected in human umbilical vein endothelial cells (HUVECs) challenged with 10–40 μM AA [36]. The endothelial Ca^2+^ response to AA can, however, be induced by many of its metabolites. For instance, 5,6-EET and 8,9-EET mediate AA-evoked Ca^2+^ response in mouse aortic endothelial cells [32,33], whereas leukotriene C4 is responsible for AA-induced Ca^2+^ influx in HUVECs [36]. Nevertheless, ETYA, a non-metabolizable analogue of AA, evoked an increase in in [Ca^2+^]_i_ in hCMEC/D3 cells whose magnitude and kinetics were similar to those induced by AA, as also observed in bovine aortic endothelial cells [21], b-TECs [37] and ECFCs [22]. It is, therefore, possible to conclude that AA directly induces intracellular Ca^2+^ signalling in hCMEC/D3 cells.

### 4.3. The Complex Mechanism of AA-Evoked Intracellular Ca^2+^ Signals in hCMEC/D3 Cells: InsP_3_R3, TPC1-2, and TRPV4

A wide repertoire of mechanisms may be exploited by AA to increase the [Ca^2+^]_i_ in endothelial cells. For instance, the Ca^2+^ response to AA is mediated by intracellular Ca^2+^ release through InsP_3_Rs and TPCs and by extracellular Ca^2+^ influx through TRPV4 in ECFCs [22], whereas TRPV4 mediates AA-evoked Ca^2+^ signals in b-TECs [43], mouse aortic [32,33] and pulmonary artery [35] endothelial cells. A preliminary report showed that AA induced both endogenous Ca^2+^ release and extracellular Ca^2+^ influx in primary human brain microvascular endothelial cells. This investigation suggested the involvement of ER Ca^2+^ release through InsP_3_Rs, but some Ca^2+^ mobilization still occurred in the presence of 2-APB. Furthermore, the pathway responsible for extracellular Ca^2+^ influx was not evaluated [23]. 

#### 4.3.1. InsP_3_R3 and TPC1-2

Herein, manipulation of extracellular Ca^2+^ levels revealed that the Ca^2+^ response to AA in hCMEC/D3 cells required both intra- and extracellular Ca^2+^ sources. AA-evoked Ca^2+^ release was strongly impaired by preventing InsP_3_R3-dependent ER Ca^2+^ mobilization with 2-APB and CPA. In agreement with these observations, it has beed demonstrated that AA directly activates InsP_3_Rs in several cellular types [52,53,54]. Besides the ER Ca^2+^ pool, AA is capable of mobilizing lysosomal Ca^2+^ through the recently identified TPC1-2, as demonstrated in rat pancreatic β-cells [38] and ECFCs [22]. Accordingly, AA-evoked intracellular Ca^2+^ release was impaired by disrupting the lysosomal Ca^2+^ reservoir with GPN and by blocking TPC1-2 with NED-19. It is not clear how AA induces lysosomal Ca^2+^ release in hCMEC/D3 cells. It has recently been demonstrated that a remarkable Ca^2+^-mediated cross-talk is established between the ER and lysosomal Ca^2+^ pools [55,56,57]. For instance, the ER Ca^2+^ mobilized by InsP_3_ may be directly transferred into the acidic vesicles at ER-lysosome contact sites, thereby increasing the lysosomal Ca^2+^ content [58]. An increase in intraluminal Ca^2+^ could in turn activate TPC2 and, sometimes, also TPC1 [40]. An alternative hypothesis, which does not rule out the previous one, implies that InsP_3_-dependent Ca^2+^ release stimulates the Ca^2+^-dependent production of nicotinic acid adenine dinucleotide phosphate (NAADP), the physiological agonist of TPCs [40,59]. Finally, TPC1-2 could trigger InsP_3_-dependent Ca^2+^ release from the ER through the Ca^2+^-induced Ca^2+^ release process [60,61]. Whatever the interaction mode, it is clear that both InsP_3_R3 and TPC1-2 contribute to AA-evoked intracellular Ca^2+^ release in hCMEC/D3 cells. 

#### 4.3.2. TRPV4

As for extracellular Ca^2+^ influx, the available evidence hints at the involvement of TRPV4. First, a functional TRPV4 protein is expressed in these cells, as testified by Western Blot analysis and the Ca^2+^ response to GSK, a selective TRPV4 agonist [47]. In addition, GSK was found to induce exclusively extracellular Ca^2+^ influx and its action was inhibited by RN-1734, a specific TRPV4 blocker. Second, RN-1734 mimicked extracellular Ca^2+^ removal, by reducing the amplitude and curtailing the Ca^2+^ response to AA. Third, AA was still able to induce RN-1734-sensitive Ca^2+^ influx upon full activation of SOCE, which represents an alternative pathway for Ca^2+^ influx in hCMEC/D3 cells [17]. In addition, AA has long been known to inhibit [18,22,24,25], rather than inducing, SOCE. Finally, Orai3, which is expressed in hCMEC/D3 cells [17] and mediates AA-induced extracellular Ca^2+^ influx in HUVECs [36] and rat carotid artery vascular smooth muscle cells [62], is not directly gated by AA, but by leukotriene C4. Conversely, extracellular Ca^2+^ influx in hCMEC/D3 cells is not promoted by AA metabolites (see Paragraph 4.1). There is, therefore, strong evidence to support the conclusion that TRPV4 is responsible for AA-induced Ca^2+^ influx in hCMEC/D3 cells.

### 4.4. AA Induces NO Release in hCMEC/D3 Cells: The Rationale for the Present Investigation

The physiological role of AA metabolites in NVC has recently been extensively reviewed [2,7]. It has been largely accepted that neuronal activity leads to PGE2 synthesis via COX-2 and COX-1 in neurons and astrocytes, respectively, to EET production via cytochrome P450 epoxygenase in glial cells, and 20-HETE synthesis in VSMCs and pericytes via cytochrome P450 4A2 [2,4,5,6]. Intriguingly, the vasomotor effect of AA metabolites is finely tuned by NO, which suppresses 20-HETE formation, thereby enabling PGE2-dependent vasodilation in the somatosensory cortex [9] and the molecular layer of the cerebellum [63]. It has, however, long been known that the pharmacological blockade of all the signalling pathways leading to an increase in CBF, i.e., NO, PGE2, EET, and K^+^ signalling, does not lead to a complete inhibition of the hemodynamic response [4]. While the molecular mechanisms driving CBF may vary along the cerebrovascular tree, the signalling mechanisms of NVC could be highly redundant in order to ensure proper blood supply to the working brain [64]. Quite unexpectedly [4], it has been recently demonstrated that the hemodynamic response may be initiated at capillary level by various mechanisms [5,10,63], including the endothelial cell ability to detect neuronal activity [13]. In addition, synaptic activity was found to induce NO-mediated vasorelaxation in penetrating arterioles and capillaries through the activation of endothelial NMDARs [11]. These in vivo findings were confirmed by the evidence, provided in vitro, that glutamate and acetylcholine were able to induce repetitive Ca^2+^ transients in mouse brain endothelial cells [14,16]. Likewise, acetylcholine-induced intracellular Ca^2+^ signals and NO release were reported in hCMEC/D3 cells [17]. 

#### 4.4.1. AA Induces Delayed NO Release in hCMEC/D3 Cells

As it has long been known that AA directly induces Ca^2+^-dependent NO release throughout peripheral circulation [18,65] and in ECFCs [22], which represent the only known truly endothelial precursors in flowing blood [66], we reasoned it was worth assessing whether this mechanism was at work also in brain microvascular endothelial cells. However, NO release started with a significantly longer latency (~150 sec) as compared to the onset of the Ca^2+^ signal, which was immediate. The delayed kinetics of the increase in DAF-FM noted in response to AA are similar to observations in mouse brain microvascular endothelial cells challenged with glutamate [14]. As discussed elsewhere [5,14,67], DAF-FM does not truly detect NO, but it rather reacts with multiple nitrogen derivatives, including ONOO, NO_2_, and N_2_O_3_. We cannot, therefore, rule out the hypothesis that the NO signal arises before the increase in DAF-FM fluorescence is detectable by our imaging system. Yet, it seems unlikely that such a delayed NO release may induce the rapid increase in CBF that follows neuronal activity with a latency of ~600 msec [68]. So, it is conceivable that the major role of AA-induced NO release from human cerebrovascular endothelial cell is other than controlling NVC. Early studies demonstrated that AA-dependent NO release stimulated angiogenesis in different types of endothelial cells, including bovine aortic endothelial cells [18], b-TECs [19,43], and ECFCs [22]. Of note, AA promotes proliferation and migration also in rat brain microvascular endothelial cells [69,70]. Future investigation is required to assess whether AA-induced NO production also stimulates angiogenesis in hCMEC/D3 cells.

#### 4.4.2. AA Induced NO Release via TPC1-2 and TRPV4 in hCMEC/D3 Cells

The mechanism whereby AA induces NO release in hCMEC/D3 cells is, however, somehow different from that reported in other endothelial cell types. For instance, TRPV4 provides the source of Ca^2+^ necessary for AA-dependent recruitment of eNOS in b-TECs [19,43], while TRPV4 is supported by InsP_3_Rs and TPC1 in ECFCs [22]. Herein, we found that AA-evoked NO production in hCMEC/D3 cells was mainly driven by TPC1-2 and TRPV4 channels. Accordingly, the increase in DAF-FM fluorescence induced by AA was significantly reduced both by removal of extracellular Ca^2+^ and by pharmacological blockade of TRPV4 with RN-1734. Likewise, the impairment of lysosomal Ca^2+^ mobilization with GPN, which disrupts acidic vesicles, or NED-19, which blocks TPC1-2, suppressed AA-evoked NO release in hCMEC/D3 cells. These observations support the emerging role of lysosomal Ca^2+^ release in driving eNOS activation throughout vascular endothelium: TPC1 and/or TPC2 were indeed found to stimulate NO production in human and rat aortic endothelial cells [71], mouse [14] and human [17] brain microvascular endothelial cells, and ECFCs [22]. Conversely, preventing InsP_3_-dependent ER Ca^2+^ release with 2-APB, which inhibits InsP_3_R3, or with CPA, which depletes the ER Ca^2+^ store [17], did not significantly affect eNOS activation by AA. This result was somehow unexpected as InsP_3_R3 supports acetylcholine-induced NO production in hCMEC/D3 cells [17]. It is, therefore, conceivable that the pool of ER-embedded InsP_3_R3 is not coupled to eNOS.

## 5. Conclusions

We provided the first evidence that AA, whose metabolites are known to modulate CBF, is *per se* able to induce Ca^2+^-dependent NO release in a human cerebrovascular endothelial cell line. The Ca^2+^ response to AA requires endogenous Ca^2+^ release through ER-embedded InsP_3_R3 and lysosomal TPC1-2 and extracellular Ca^2+^ influx through TRPV4. However, only TPC1-2 and TRPV4 drive NO production, while InsP_3_R3 plays no sizeable role in this signal. This finding suggests that, besides being metabolized into the vasorelaxing mediators PGE2 and EETs, AA could induce NO release from human brain microvascular endothelial cells. It is, however, unlikely that AA-induced endothelial-dependent NO release is directly involved in the onset of the hemodynamic response to neuronal activity. These findings should, however, be confirmed in vivo by taking advantage of the availability of mouse models expressing endothelial genetic Ca^2+^ indicators [72]. However, a recent report confirmed that acetylcholine was able to induce endothelial NO release in vivo [73], as suggested by previous in vitro observations in hCMEC/D3 cells [17]. This same report provided the evidence that also synaptically released glutamate induced endothelial NO production [73], as previously observed in a mouse brain cerebrovascular endothelial cell line [14].

## Figures and Tables

**Figure 1 cells-08-00689-f001:**
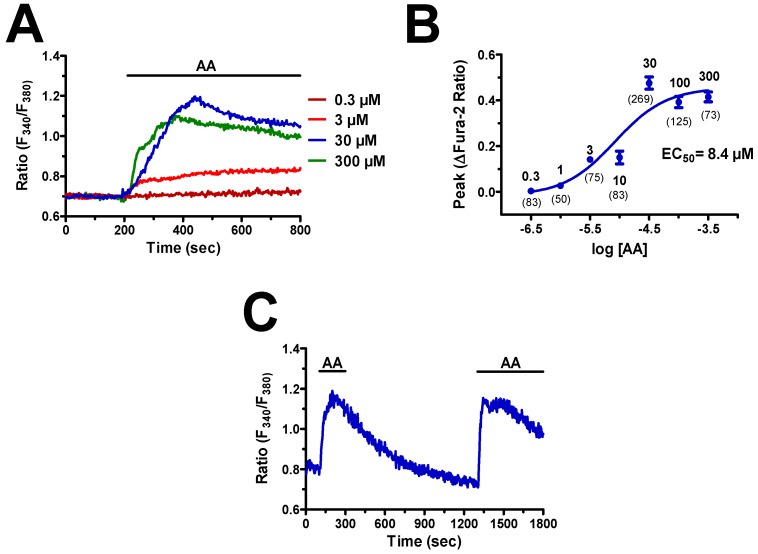
Arachidonic acid (AA) induces an increase in intracellular Ca^2+^ concentration ([Ca^2+^]_i_) in hCMEC/D3 cells. (**A**) Intracellular Ca^2+^ signals evoked by increasing doses of AA in the hCMEC/D3 cell line. In this and the following figures, agonists and drugs were administered as indicated by the horizontal bars above the traces. (**B**) Dose–response relationship of the amplitude of AA-evoked Ca^2+^ signals in hCMEC/D3 cells. The continuous line was obtained from a fit to a sigmoidal concentration-response curve by using Equation (1). (**C**) 30 μM AA evoked a rapid increase in [Ca^2+^]_i_ that decayed to the baseline upon agonist removal from the perfusate. When AA was re-added at the same dose, it induced a second elevation in [Ca^2+^]_i_ that achieved the same peak amplitude as the first one.

**Figure 2 cells-08-00689-f002:**
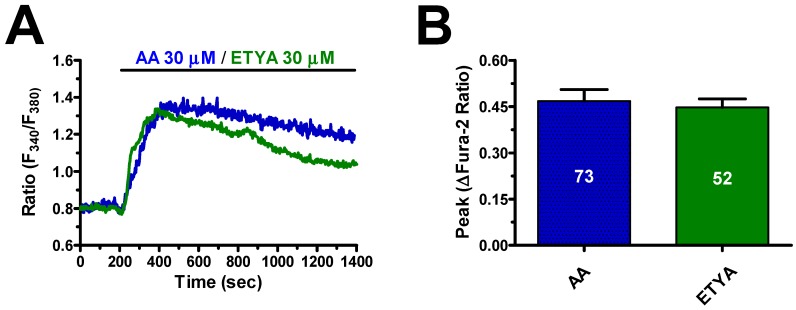
Arachidonic acid (AA) metabolism is not necessary for arachidonic acid-induced elevation in [Ca^2+^]_i_ in hCMEC/D3 cells. (**A**) 30 μM eicosatetraynoic acid (ETYA), a non-metabolizable analogue of AA, elicits an intracellular Ca^2+^ signal which overlaps with the Ca^2+^ response to 30 μM AA in hCMEC/D3 cells. (**B**) Mean ± SE of the amplitude of ETYA- and AA-induced elevations in [Ca^2+^]_i_.

**Figure 3 cells-08-00689-f003:**
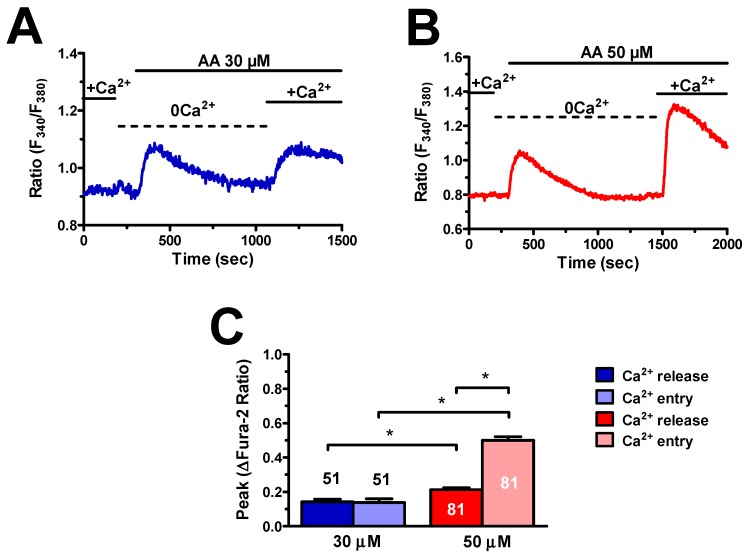
Arachidonic acid induces endogenous Ca^2+^ release and extracellular Ca^2+^ influx in hCMEC/D3 cells. (**A**) 30 μM AA evoked endogenous Ca^2+^ release under absence of extracellular Ca^2+^ (0Ca^2+^) conditions followed by a second increase in [Ca^2+^]_i_ upon Ca^2+^ restitution to the bath in the continuous presence of the agonist. (**B**) 50 μM AA evoked endogenous Ca^2+^ release under 0Ca^2+^ conditions followed by a second increase in [Ca^2+^]_i_ upon Ca^2+^ restitution to the bath in the continuous presence of the agonist. (**C**) Mean ± SE of the amplitude of endogenous Ca^2+^ release and extracellular Ca^2+^ influx induced by each concentration of AA. The asterisk indicates *p* < 0.05.

**Figure 4 cells-08-00689-f004:**
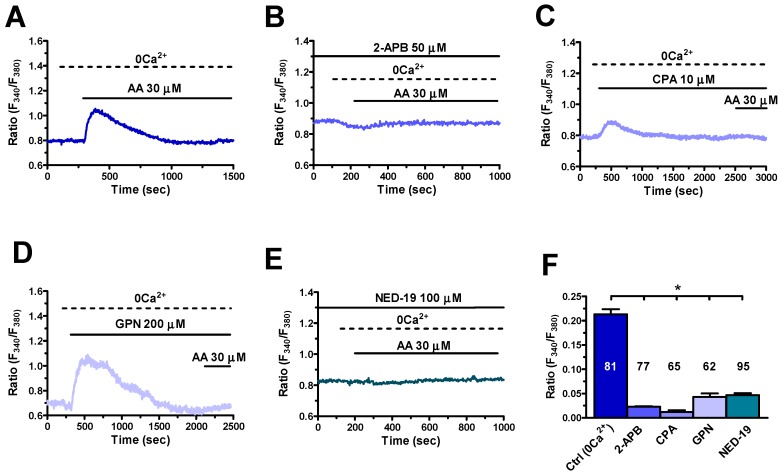
Type 3 inositol-1,4,5-trisphosphate receptor 3 InsP_3_R3 and two-pore channels 1 and 2 (TPC1-2) mediate endogenous Ca^2+^ release in response to arachidonic acid in hCMEC/D3 cells. (**A**) Endogenous Ca^2+^ mobilization induced by AA (30 μM) under 0Ca^2+^ conditions. (**B**) 2-aminoethoxydiphenyl borate (2-APB; 50 μM, 30 min) prevents AA-evoked endogenous Ca^2+^ release. In this and in the following panels, AA has always been applied at 30 μM. (**C**) Cyclopiazonic acid (CPA; 10 μM) inhibited Sarco-Endoplasmic Reticulum Ca^2+^-ATPase (SERCA) activity, thereby eliciting a transient elevation in [Ca^2+^]_i_ followed by Ca^2+^ removal from the cytosol. The following addition of AA did not result in an increase in [Ca^2+^]_i_. (**D**) Glycyl-L-phenylalanine 2-naphthylamide (GPN; 200 μM) elicited a transient elevation in [Ca^2+^]_i_ due to the disruption of the lysosomal Ca^2+^ pool and the following liberation of lysosomal Ca^2+^ content into the cytosol [41,42]. The following addition of AA did not result in an increase in [Ca^2+^]_i_. (**E**) NED-19) (100 μM, 30 min) blocked AA-evoked intracellular Ca^2+^ release. (**F**) Mean ± SE of the amplitude of the intracellular Ca^2+^ response to AA under the designated treatments. The asterisk indicates *p* < 0.05.

**Figure 5 cells-08-00689-f005:**
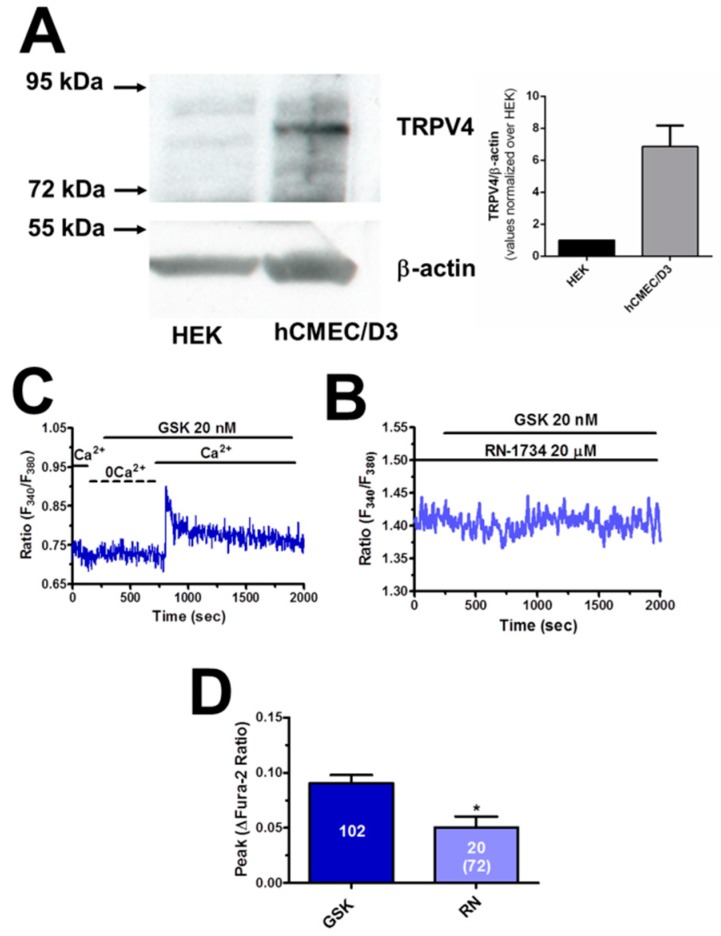
Transient Receptor Potential Vanilloid 4 (TRPV4) protein is expressed and mediates extracellular Ca^2+^ influx in hCMEC/D3 cells. (**A**) Left, representative Western blots illustrating TRPV4 and β-actin expression in HEK-293 cells (HEK) and hCMEC/D3 cells. Left, quantification of TRPV4/β-actin protein levels in hCMEC/D3 cells normalized on HEK cells from three independent experiments. (**B**) GSK1016790A (GSK; 20 nM) increased the [Ca^2+^]_i_ in the presence, but not in the absence (0Ca^2+^), of extracellular Ca^2+^. (**C**) RN-1734 (20 μM, 20 min) blocked GSK-evoked extracellular Ca^2+^ influx. GSK was administered at 20 nM. (**D**) Mean ± SE of the amplitude of GSK-induced Ca^2+^ influx in the absence and in the presence of RN-1734 (RN). The total number of cells analysed in the presence of RN is indicated in the brackets. The asterisk indicates *p* < 0.05.

**Figure 6 cells-08-00689-f006:**
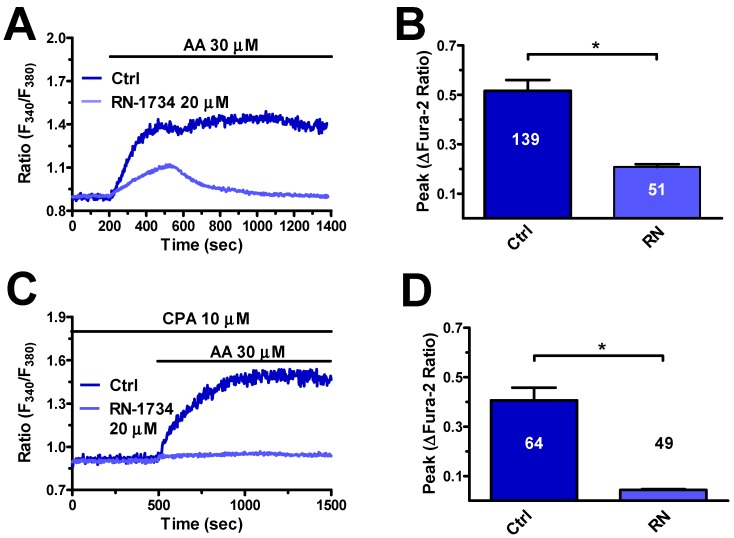
TRPV4 mediates arachidonic acid-induced extracellular Ca^2+^ influx in hCMEC/D3 cells. (**A**) Intracellular Ca^2+^ response to AA (30 μM) recorded in the presence and in the absence of RN-1734 (20 μM, 20 min). (**B**) Mean ± SE of the amplitude of AA-evoked Ca^2+^ influx in the absence and in the presence of RN-1734 (RN). The asterisk indicates *p* < 0.05. (**C**) AA (30 μM) evoked an increase in the [Ca^2+^]_i_ in the presence of extracellular Ca^2+^ upon 30 min treatment with CPA (10 μM) to deplete the ER Ca^2+^ store and fully activate store-operated Ca^2+^ influx (SOCE). AA-evoked Ca^2+^ influx was abolished by pretreating the cells with RN-1734 (20 μM, 20 min). (**D**) Mean ± SE of the amplitude of AA-induced intracellular Ca^2+^ signals in the absence and in the presence of RN-1734 (RN) upon SOCE activation with CPA (10 μM). The asterisk indicates *p* < 0.05.

**Figure 7 cells-08-00689-f007:**
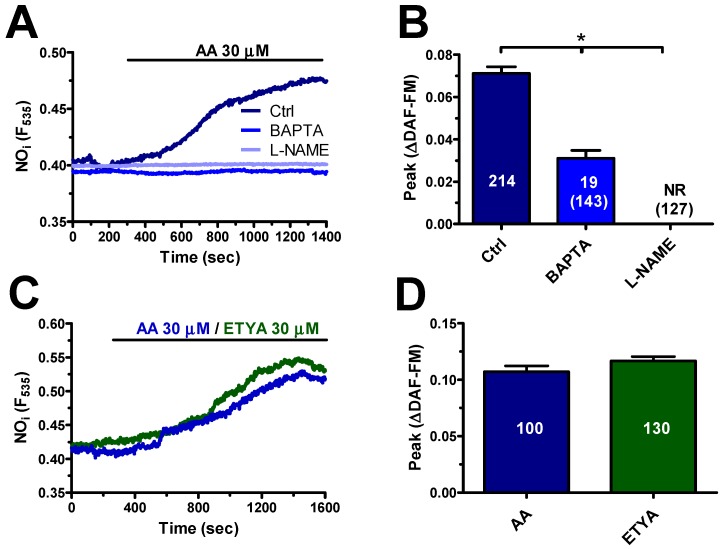
Arachidonic acid induces NO release in a Ca^2+^-dependent manner in hCMEC/D3 cells. (**A**) AA (20 μM) evoked a robust increase in the fluorescence in hCMEC/D3 cells loaded with the NO-selective fluorophore, 4-amino-5-methylamino-2’,7’-difluorofluorescein (DAF-FM). AA-elicited NO release was abrogated by L-N^G^-Nitro-L-arginine methyl ester (L-NAME; 100 μM, 1 h) or BAPTA (30 μM, 2 h). (**B**) Mean ± SE of the amplitude of AA-elicited NO release under the designated treatments. For L-NAME and BAPTA, the number of total cells analysed is indicated in the brackets. The asterisk indicates *p* < 0.05. (**C**) AA (30 μM) and ETYA (30 μM) induced an increase in DAF-FM fluorescence with overlapping kinetics and amplitude. (**D**) Mean ± SE of the amplitude of AA- and ETYA-evoked NO release in hCMEC/D3 cells. NR means no response.

**Figure 8 cells-08-00689-f008:**
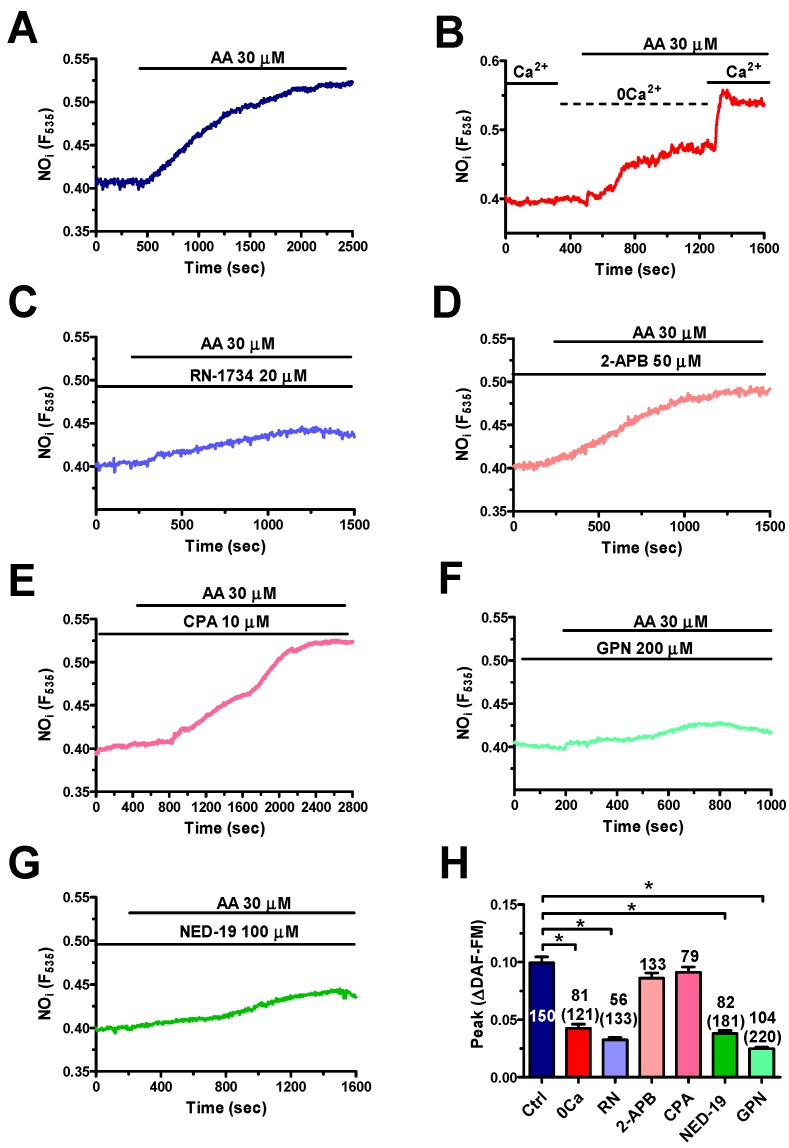
TPC1-2 and TRPV4 support arachidonic acid-induced NO production in hCMEC/D3 cells. (**A**) AA (30 μM) evoked robust NO release under control conditions in hCMEC/D3 cells. (**B**) AA (30 μM) induced NO release under 0Ca^2+^ conditions, although this signal was lower as compared to control conditions. However, readdition of Ca^2+^ to the perfusate caused a second increase in DAF-FM fluorescence. (**C**) RN-1734 (20 μM, 20 min) reduced AA-evoked NO release. AA was administered at 30 μM. (**D**) 2-APB (50 μM, 30 min) did not affect AA-induced NO production. AA was administered at 30 μM. (**E**) CPA (10 μM, 30 min) did not impair AA-evoked NO release. AA was administered at 30 μM. (**F**) GPN (200 μM, 30 min) remarkably reduced AA-induced NO production. AA was administered at 30 μM. (**G**) NED-19 (100 μM, 30 min) impaired AA-evoked NO release. AA was administered at 30 μM. (**H**) Mean ± SE of the amplitude of AA-evoked NO release under the designated treatments. The asterisk indicates *p* < 0.05. The total number of analysed cells is indicated in the brackets.

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
