# Peer review of "Arachidonic Acid Evokes an Increase in Intracellular Ca2+ Concentration and Nitric Oxide Production in Endothelial Cells from Human Brain Microcirculation"

_cells, 2019, doi:10.3390/cells8070689_

Round 1
Reviewer 1 Report
Arachidonic acid (AA) is a fatty acid that can be released from membrane phospholipids by the action of certain phospholipases. AA itself has many effects on cells; in addition, it can be converted into other active intermediates. AA also has an important role in mediating changes in local cerebral blood flow in response to neuronal activity. The authors of this interesting and detailed investigation used a human brain microvascular endothelial cell line (hCMEC/D3) to examine the effects of exogenous AA on intracellular concentrations of Ca2+ and on the release of nitric oxide. The AA-mediated release of nitric oxide was dependent on Ca2+, and the mechanisms responsible for AA-induced changes in intracellular Ca2+ concentrations were delineated. Because the rate of AA-induced nitric oxide release was relatively slow, however, the authors judged it unlikely to account for the ability of AA to quickly increase local cerebral blood flow in response to neuronal activity.
The authors did a great deal of work in this very thorough study, and the data are generally presented very well. The experiments were carefully designed and the conclusions are reasonable.
There are, however, some aspects of the manuscript that could be clarified and improved. The reviewer respectfully offers the following comments for consideration by the authors. Comments are generally listed in order of their appearance in the manuscript, rather than in their order of importance. The reviewer’s last comment is more important than many of the others.
Abstract, line 40. The abbreviation NVC is used in line 40 before it is defined in line 64.
Figure 1B. The reviewer assumed that the numbers in parentheses on the the graph represent the numbers of cells observed. The authors may want to mention this point in the figure legend.
Line 255. In the sentence that begins, “Of note, a throughout characterization….”, the authors probably meant to state, “Of note, a thorough characterization….”.
Figure 7B. The reviewer assumed that the abbreviation NR meant “None Released”. To improve clarity, the authors may wish to state the meaning of this abbreviation in the figure legend.
Lines 370 & 371. In the sentence beginning, “Conversely, preventing lysosomal Ca2+ release…….”, the first graph referenced should be Figure 8G, and the second graph referenced should be Figure 8F.
Figure 8. The size of Figure 8 should be increased. In its current form, the different sections of Figure 8 are too small to be easily read.
Line 449. The abbreviation “CICR” should be defined.
Line 480. Shouldn’t this read, “…of endothelial NMDARs (11).”?
Lines 485-486. Shouldn’t this read, “…we reasoned that it was worth assessing whether….”?
Line 488. In the reviewer’s opinion, this part would read better as “…the Ca2+ signal, which was immediate.”
Lines 488-490. Below is the reviewer’s suggested revision of the sentence that begins with, “The delayed kinetics….”.
“The delayed kinetics of the increase in DAF-FM fluorescence noted in response to AA are similar to observations in mouse brain microvascular endothelial cells challenged with glutamate (14)”
Lines 505-506. The following sentence needs to be revised: “Herein, we found that TPC1-2 and TRPV4 are the main responsible for AA-evoked NO release in hCMEC/D3 cells.”
Line 510. Since there is more than one observation, the reviewer suggests that this sentence begin as follows: “These observations support the emerging role….”
Line 513-515. The reviewer has two suggestions for the following sentence, “Conversely, preventing InsP3-dependent ER Ca2+ release with either U73122, to block PLC activation, or 2APB, to inhibit InsP3R3 did not significantly affect eNOS activation by AA.” First, although it is probably familiar to most readers, the abbreviation “PLC” should be defined. Second, the reviewer suggests that a reference be cited at the end of this sentence.
Line 524. Suggested revision: “…besides being metabolized into the vasorelaxing mediators…”
General comment about the Discussion section. Each section of the Discussion section in this manuscript generally consists of a single, very lengthy paragraph. In some instances, a given section would be much easier to follow if it were divided clearly into multiple paragraphs that emphasize the main points.
Author Response
Dear Referee #1,
We were very grateful for your nice comments to our manuscript entitled "Arachidonic acid evokes an increase in intracellular Ca2+ concentration and nitric oxide production in endothelial cells from human brain microcirculation” recently submitted for publication as Research Article in Cells – Special Issue Phospholipids: Dynamic Lipid Signaling in Health and Diseases.
We truly believe that your comments did improve the quality of this manuscript and we are grateful for the attention you paid to our work. We amended the manuscript according to your indications and addressed all the criticisms you raised. More specifically:
Abstract, line 40. The abbreviation NVC is used in line 40 before it is defined in line 64.
We thank the referee for this observation. We amended this inaccuracy.
Figure 1B. The reviewer assumed that the numbers in parentheses on the graph represent the numbers of cells observed. The authors may want to mention this point in the figure legend.
This is correct. Actually, we originally specified this in the Methods, Paragraph 2.5 (Statistical analysis). If the Reviewer prefers this information to be moved in the Figure legends, we will be glad to follow his/her advice.
Line 255. In the sentence that begins, “Of note, a throughout characterization….”, the authors probably meant to state, “Of note, a thorough characterization….”.
We thank the Reviewer for this elegant comment. The Reviewer is definitively right.
Figure 7B. The reviewer assumed that the abbreviation NR meant “None Released”. To improve clarity, the authors may wish to state the meaning of this abbreviation in the figure legend.
The Reviewer is right and we think for this observation. The abbreviation NR means No Response, a king of None Release, and we specified this in the Figure legend.
Lines 370 & 371. In the sentence beginning, “Conversely, preventing lysosomal Ca2+ release…….”, the first graph referenced should be Figure 8G, and the second graph referenced should be Figure 8F.
The Reviewer is definitively right. We amended the text accordingly.
Figure 8. The size of Figure 8 should be increased. In its current form, the different sections of Figure 8 are too small to be easily read.
We thank the Reviewer for this observation. We increased the size of Figure 8.
Line 449. The abbreviation “CICR” should be defined.
The Reviewer is right. The abbreviation, which means Ca2+-induced Ca2+ release, has been defined.
Line 480. Shouldn’t this read, “…of endothelial NMDARs (11).”?
The Reviewer is right. We amended this typo.
Lines 485-486. Shouldn’t this read, “…we reasoned that it was worth assessing whether….”?
The Reviewer is again fully right. We amended the sentence according to his/her advice.
Line 488. In the reviewer’s opinion, this part would read better as “…the Ca2+ signal, which was immediate.”
We thank the Reviewer for this comment and we changed the text accordingly.
Lines 488-490. Below is the reviewer’s suggested revision of the sentence that begins with, “The delayed kinetics….”.
“The delayed kinetics of the increase in DAF-FM fluorescence noted in response to AA are similar to observations in mouse brain microvascular endothelial cells challenged with glutamate (14)”
We again thank the Reviewer for this elegant comment and we changed the text accordingly.
Lines 505-506. The following sentence needs to be revised: “Herein, we found that TPC1-2 and TRPV4 are the main responsible for AA-evoked NO release in hCMEC/D3 cells.”
We thank the Reviewer for this observation and changed the text in: “Herein, we found that AA-evoked NO release in hCMEC/D3 cells is mainly driven by TPC1-2 and TRPV4 channels”.
Line 510. Since there is more than one observation, the reviewer suggests that this sentence begin as follows: “These observations support the emerging role….”
We thank the reviewer for this observation and changed the text accordingly.
Line 513-515. The reviewer has two suggestions for the following sentence, “Conversely, preventing InsP3-dependent ER Ca2+ release with either U73122, to block PLC activation, or 2APB, to inhibit InsP3R3 did not significantly affect eNOS activation by AA.” First, although it is probably familiar to most readers, the abbreviation “PLC” should be defined. Second, the reviewer suggests that a reference be cited at the end of this sentence.
We thank the Reviewer for these suggestions that we were glad to follow. Actually, we replaced PLC (which means phospholipase C, as the Reviewer has certainly argued) with CPA, which is is the drug that we used in our experiments.
Line 524. Suggested revision: “…besides being metabolized into the vasorelaxing mediators…”
Again, we thank the Reviewer for this elegant comment.
General comment about the Discussion section. Each section of the Discussion section in this manuscript generally consists of a single, very lengthy paragraph. In some instances, a given section would be much easier to follow if it were divided clearly into multiple paragraphs that emphasize the main points.
We thank the Reviewer for this important observation. Each paragraph has been clearly divided in multiple subparagraphs to emphasize the main points as kindly suggested by the Reviewer.
We hope that you will now consider our manuscript worth of being published on this important Special Issue of Cells.
Reviewer 2 Report
TO THE AUTHORS:
Cells; MS# 542825
Title: Arachidonic acid evokes an increase in intracellular Ca2+ concentration and nitric oxide production in endothelial cells from human brain microcirculation
Berra-Romani et al wish to publish a manuscript elucidating the mechanism by which arachidonic acid elicits sequential calcium and nitric oxide signaling in a human brain microvascular endothelial cell line. The authors primarily use fluorescent dye-based photometry (Fura-2 for calcium & DAF-FM for nitric oxide) in combination with pharmacology for the endoplasmic reticulum inositol-1,4,5-trisphosphate receptors, lysosomal two-pore channels, and the plasma membrane transient receptor vanilloid (type 4) channels. Overall, the manuscript is written and presented well throughout from premise/experimental objective to a reasonably thorough discussion of findings. See concerns below in order of importance starting from the greatest to the least:
(1) A major concern is the abundance of several similar manuscripts (from the same group; e.g., see PMIDs 30451297, 30191989, 28807148, 27634591). It is not understood why the current study (or the others) couldn’t have been combined in scope and experimentation to allow for more complete reports. Does the current manuscript include any significantly overlapping text/data with previously published studies? If so, appropriate copyright permissions should be obtained accordingly and cited in the manuscript. Regardless, the publication record of the group appears to be a collective effort towards maximizing the number of publications vs. optimizing quality and completeness per study and its respective manuscript. This practice is generally not accepted well by the broader scientific community.
(2) Two major experimental limitations include use of a cultured cell line (vs. primary isolation) and an experimental temperature at room (vs. physiological, 37°C). Both factors significantly alter physiological conditions vs. mammalian conditions in vivo. It is the reviewer’s opinion that these limitations should be mentioned in the Discussion with some speculation as to whether their key conclusions remain the same accordingly.
(3) Indicate whether the blot shown in Figure 5A is a representation of multiple blots or not. As an additional panel, can the authors show the summary data for the Western Blots as a densitometry comparison +/- standard error of the mean?
(4) Figures 1B, 2B, 3C, 4F, 5D, 6B&D, and 8H should indicate the change or “delta” in the raw fluorescent signal on the y-axis, not raw fluorescence itself as accurately indicated for the experimental traces. The respect Figure Legends corresponding to those same panels should also clearly state what the numbers shown in or around the summary data mean (e.g., n-value of cell numbers). As inconsistent with the other summary data, there are no numbers indicated for Figure 4F. Finally, all panels for Figure 8 should be enlarged by at least 2 to 3 times as they appear too small to readily read, examine, and interpret.
(5) The authors could reduce the use of some unnecessary abbreviations such as “NA” for neuronal activity which could be confused for “noradrenaline” for example.
Author Response
Dear Referee #2,
We were very grateful for your nice comments to our manuscript entitled "Arachidonic acid evokes an increase in intracellular Ca2+ concentration and nitric oxide production in endothelial cells from human brain microcirculation” recently submitted for publication as Research Article in Cells – Special Issue Phospholipids: Dynamic Lipid Signaling in Health and Diseases.
We truly believe that your comments did improve the quality of this manuscript and we are grateful for the attention you paid to our work. We amended the manuscript according to your indications and addressed all the criticisms you raised. More specifically:
(1) A major concern is the abundance of several similar manuscripts (from the same group; e.g., see PMIDs 30451297, 30191989, 28807148, 27634591). It is not understood why the current study (or the others) couldn’t have been combined in scope and experimentation to allow for more complete reports. Does the current manuscript include any significantly overlapping text/data with previously published studies? If so, appropriate copyright permissions should be obtained accordingly and cited in the manuscript. Regardless, the publication record of the group appears to be a collective effort towards maximizing the number of publications vs. optimizing quality and completeness per study and its respective manuscript. This practice is generally not accepted well by the broader scientific community.
We respectfully, but firmly (very firmly), disagree with the Referee. Our group has been investigating the role of endothelial Ca2+ signaling in neurovascular coupling since three years, during which we have focused on various aspects of this topic. Actually, vascular endothelial cells react to a multitude of autacoids and neurotransmitters and, honestly, I do not remember any single paper dealing with the Ca2+/NO response of a given endothelial cell type to all the chemical stimuli it could be exposed to (e.g. histamine, thrombin, acetylcholine, bradykinin, ATP, UTP, and so on). We are carefully dissecting how brain microvascular endothelial cells regulate neurovascular coupling by examining the effect of specific neurotransmitters on Ca2+ signaling and NO release. This task takes time and detailed investigation of all the signalling pathways potentially involved. Just consider that, when hCMEC/D3 cells are stimulated with acetylcholine, SOCE is the driver of eNOS activation, while TPC1-2 and TRPV4 channels induce NO release when the same cells are challenged with arachidonic acid, as in the present investigation. I am not sure that a peer-reviewed journal would accept a paper with the 40 figures or so required to assess these multiple issues (i.e. acetylcholine, glutamate, and arachidonic acid). I would also like to recall you that Anderson and coworkers published four separate (beautiful) papers to demonstrate the occurrence of one single process, i.e. that endothelial NMDA receptors drive brain arteriole dilation in response to neuronal activity. Likewise, Hamel’s group is carefully investigating which neuronal populations drive neurovascular coupling and how they accomplish this goal. This renowned group dedicated many interesting papers to this topic. I am wondering whether Anderson’s and Hamel’s groups were also supposed to compact all the information in one single manuscript (which was impossible, due to the complexity of the topic) and published so many independent papers only to maximize the number of publications. When different aspects of the same topic are addressed, as in our case, compacting the information could only be detrimental to the quality of the research. Therefore, we strongly reject your comment that “the publication record of the group appears to be a collective effort towards maximizing the number of publications vs. optimizing quality and completeness per study and its respective manuscript” and wonder whether you also aim at publishing Your most complex results as a single manuscript.
(2) Two major experimental limitations include use of a cultured cell line (vs. primary isolation) and an experimental temperature at room (vs. physiological, 37°C). Both factors significantly alter physiological conditions vs. mammalian conditions in vivo. It is the reviewer’s opinion that these limitations should be mentioned in the Discussion with some speculation as to whether their key conclusions remain the same accordingly.
We thank the Reviewer for this observation. We have a rather long-lasting experience with Fura-2 measurements and what we learnt is that, although different cell types must be individually evaluated in terms of loading strategies (Fura-2 concentration and incubation time), a few rules almost always apply. Indeed, while it is commonly accepted that incubating the cells at 37°C for a short period facilitates dye uptake into the cells, prolonging the exposure to this temperature hampers the measurement of Ca2+ signals. This issue has been object of intense investigation in the past and, also for our experience, when the recording is carried out at 37°C the measurements may not be reliable. As discussed in several remarkable papers (Bootman et al., Cold Spring Harb Protoc 2013;2013(2):83-99; Di Virgilio et al., Biochem J 1988;256(3):959-63; Malgaroli et al., J Cell Biol 1987;105(5):2145-55; Thomas et al., Cell Calcium 2000;28(4):213-23), maintaining the temperature at 37°C hampers Ca2+ measurement as the dye can be accumulated within endogenous organelles (mainly ER and mitochondria) or be extruded across the plasma membrane through organic anion transporters. For these reasons, most colleagues prefer measuring Ca2+ at room temperature (RT). Additionally, Bootman and coworkers (Cold Spring Harb Protoc 2013;2013(2):83-99) recently discussed that “although Ca2+ signaling systems are in part enzymatic and may be affected by temperature, cellular Ca2+ signaling systems can generally be faithfully observed below 37°C”. We added this last reference to the Materials and methods, on Line 140.
As to the in vivo relevance of the findings illustrated in the present investigation, we do agree with the Reviewer’s comment. We addressed this issue in Paragraph 5 (Lines 533-538) by recalling that a recent in vivo investigation confirmed previous observations deriving from cultured cell line.
(3) Indicate whether the blot shown in Figure 5A is a representation of multiple blots or not. As an additional panel, can the authors show the summary data for the Western Blots as a densitometry comparison +/- standard error of the mean?
We thank the Reviewer for this observation. The blot is representative of three independent experiments and the summary data were shown as a densitometry comparison in the revised figure.
(4) Figures 1B, 2B, 3C, 4F, 5D, 6B&D, and 8H should indicate the change or “delta” in the raw fluorescent signal on the y-axis, not raw fluorescence itself as accurately indicated for the experimental traces. The respect Figure Legends corresponding to those same panels should also clearly state what the numbers shown in or around the summary data mean (e.g., n-value of cell numbers). As inconsistent with the other summary data, there are no numbers indicated for Figure 4F. Finally, all panels for Figure 8 should be enlarged by at least 2 to 3 times as they appear too small to readily read, examine, and interpret.
We thank the Reviewer for these observations. We changed the Y-axis labels where required and added the number of experiments in Figure 4F. In Materials and methods (Lines 174-175) we specified that the number in the bar histograms indicate the number of analyzed cells. Finally, Figure 8 was enlarged as suggested by the Reviewer.
(5) The authors could reduce the use of some unnecessary abbreviations such as “NA” for neuronal activity which could be confused for “noradrenaline” for example.
We thank the Reviewer for this observation. All the unnecessary abbreviations, including “NA” were eliminated.
We hope that the Reviewer will now consider our manuscript worth of being published on this very important Special Issue of Cells.
Round 2
Reviewer 2 Report
I apologize for the apparently provocative nature of my comment #1. I am just trying to deliver a thorough (and honest) job as your peer and colleague in light of the anticipated readership of the manuscript. There are varying philosophies and abilities inherent to each laboratory as to how much data could (or should) be included in a single manuscript for publication. I will not debate this topic in depth any further but do feel that the authors could have addressed this concern in a sentence or two. I definitely don't recommend responding with long, exaggerated rants for your reviewers in the future. Of course, no one is asking for a 40-figure paper with every endothelial cell agent in existence tested. If the experienced reviewer did not feel that potential overlap among studies (or the perception thereof) was going to be a valid concern among the future readership, the reviewer would not have approached the authors with it.